# Efficacy of Visual Feedback Training for Motor Recovery in Post-Operative Subjects with Knee Replacement: A Randomized Controlled Trial

**DOI:** 10.3390/jcm11247355

**Published:** 2022-12-11

**Authors:** Simone Carozzo, Martina Vatrano, Francesco Coschignano, Riccardo Battaglia, Rocco Salvatore Calabrò, Loris Pignolo, Marianna Contrada, Paolo Tonin, Antonio Cerasa, Andrea Demeco

**Affiliations:** 1S’Anna Institute, 88900 Crotone, Italy; 2IRCCS Centro Neurolesi “Bonino Pulejo”, 98124 Messina, Italy; 3Institute for Biomedical Research and Innovation (IRIB), National Research Council of Italy, 98164 Messina, Italy; 4Pharmacotechnology Documentation and Transfer Unit, Preclinical and Translational Pharmacology, Department of Pharmacy, Health Science and Nutrition, University of Calabria, 87036 Rende, Italy

**Keywords:** kinematic parameters, motor recovery, total knee replacement, visual feedback

## Abstract

To evaluate the effects of visual feedback training on motor recovery in postoperative patients with a total knee replacement (TKR). The performance of 40 first-ever TKR patients (27 females; mean age: 70.5 (67.2–74.0) years) was evaluated in a single center, single-blind, randomized controlled study. The patients were randomly and equally distributed into two demographically/clinically matched groups undergoing experimental or traditional treatments. All patients have been treated in a 1 h session, 2/day for 5 days a week, for six consecutive weeks. The first group (“control”) underwent conventional physical therapy, whereas the experimental group received advanced knee training with visual feedback using the TecnoBody® device (Walker View 3.0 SCX, Dalmine (BG), Italy). The clinical scales and kinematic parameters coming from the gait analysis were evaluated to demonstrate the dynamic balance function in a standing position before and after each treatment. After the treatment, both experimental and control groups improved significantly and similarly, as measured by the clinical scales (Numeric Rating Scale for Pain and Barthel index). A significant boosting of the motor performance was detected in the experimental group with respect to the control group in the terms of symmetry index 84 (80.8–85.4) vs. 87.15 (84–92.8) *p* = 0.001 *; single stance support 34.9 (34.1–36.5) vs. 37.8 (36.6–38.9); *p* < 0.001; and obliquity parameters 58.65 (51.3–70.3) vs. 73 (62.3–82.1); *p* < 0.001. Applying visual feedback training in addition to traditional rehabilitation strategies improves the knee function and motor control in postoperative TKR patients.

## 1. Introduction

For patients with severe knee osteoarthritis (OA), which affects about one in ten people over the age of 50, total knee replacement (TKR) surgery is seen to be an effective treatment [1]. A TKR is one of the most common replacement surgeries, with about 650 thousand surgeries done in the USA every year [2]. Patients usually complain of persistent pain and a subsequent reduction in their mobility, either at night or at weight bearing, which has not been relieved after six months of conservative treatments [2]. Indeed, the main outcome of a TKR is to relieve pain and improve the function of the knee in people with severe symptoms and end-stage radiographic disease [2]. However, despite a significant improvement in the symptoms, a TKR did not completely restore the knee’s functions, and the treated knee could still have biomechanical impairments, particularly when walking. The question of whether postoperative deambulation deficits are brought on by the patient’s antalgic gait pattern prior to surgery, the results of the surgical operation, or a combination of the two elements is still open for debate [3,4]. Nevertheless, there is an association between the gait alterations and muscle weakness, in particular the quadriceps, which is usually compensated by an adaptation in the ankle and hip joint [3]. On the other hand, there is a significant percentage of patients that present poor long-term outcomes after a TKR. Judge et al. [5] showed that 14 to 36% of patients do not see a functional improvement one year following surgery. Additionally, only 50% of patients report a significant improvement in their knee arthritis severity using the Western Ontario and McMaster Universities OA Index (WOMAC)score [6], and 10–34% claim that their post-operative pain ratings did not improve [7].

Thus, there is a clear need for a structured rehabilitation approach to increase recovery and target patients with unfavorable outcomes after a TKR, [8]. Inpatient facilities are thought to be a crucial component of post-surgery care for patients with a TKR. Reducing edema and boosting the range of motion and strength are critical [9] during the first year following a TKA, which is also the most crucial time to correct the gait deficiency [3]. A thorough evaluation of the patient serves as the foundation for the rehabilitation intervention. This is often done through a clinical examination and clinical scales, such as the Visual Analogue Scale for a pain assessment, and the Barthel index for the activity of the daily living assessment [10,11]. However, clinical scales are often subjective and could be influenced by a patients’ satisfaction. [12,13] Gait analysis, instead, provides an objective assessment of motion pattern changes during walking in patients with a TKR, as well as the improvement in the implant function through the investigation of the materials and design [4,14]. Bączkowicz [4] et al. reported a gait impairment which was showed through a pressure-sensitive walkway analysis in single support time, before a TKR and the step length after a TKR, with a partial restoration observed 12 weeks post-surgery and the completion of the rehabilitation plan. For this reason, reliable clinical markers of motor impairments during walking are fundamental to guide physicians in defining tailored rehabilitation programs, including workouts targeting patients’ compensatory mechanisms which are finalized to maximize the return to normal daily activities [15].

There are several approaches in post-acute settings for patients with TKR, including but not limited to exercise therapy, hydrokinesitherapy, cryotherapy, transcutaneous electrical nerve stimulation and neuromuscular electrical stimulation. Exercise therapy commonly includes passive mobilization, strength, and gait training. Even if there is no consensus on “standard” rehabilitation, the main outcomes are maximizing the knee’s range of movements (ROM)., improving the muscle power (knee stabilizer muscles), controlling the pain, and normalizing the gait [16]. Hydrokinesitherapy gains even more attractiveness, especially in early rehabilitation, thanks to the effect of water in reducing gravity, and to the water speed-dependent resistance that enhances the muscle strength [16]. Balance training is an effective method to restore the joint proprioception and postural control deficit caused by mechanoceptors damage due to surgery [16]. Furthermore, cold and compression have beneficial effects in reducing swelling and inflammation, slowing the metabolic activity of soft tissues [17], and in patients with deficits in voluntary muscle contraction, the use of neuromuscular electrical stimulation could represent a therapeutic option to limit the loss of muscle fiber due to immobilization [16]. 

In this context, visual feedback (VFB) has shown interesting results in enhancing the ability of patients in controlling the dynamic stability of the body [18,19,20]. The emerging rehabilitative technologies allow monitoring, through a force platform, and the displacement of the patient’s center of balance during proprioceptive care and gait training, providing feedback on the exact position on the computer screen. As a result, the patient gets access to multisensory information and can self-correct his posture while performing the suggested exercises. This kind of method is built upon theories of motor learning that are required for the central nervous system to act efficiently [21]. In the last twenty years, much evidence has demonstrated the effectiveness of VFB for increasing the impact of gait recovery approaches [22], although this has been mostly applied to neurologic patients. Moreover, the VFB approach has been made through equipment not intended for rehabilitation (i.e., a Wii) with limited clinical translation). Finally, another caveat is that the evaluation of the motor gain in patients after a TKR has largely been analyzed focusing on the clinical (i.e., WOMAC and VAS) or activity (Timed Up and Go Test, Five Times Sit to Stand Test,, and gait speed) scores [23].

For this reason, this study is aimed at evaluating, for the first time, the effectiveness of an advanced motor rehabilitation protocol, integrating an innovative and user-friendly VFB system in patients after a TKR, through a detailed and objective kinematic gait analysis. For a deeper gait evaluation of the patient outcomes (in terms of the motor speed, balance, and control), an Inertial measurement unit (IMU) was also used.

## 2. Materials and Methods

### 2.1. Participants

The patients were recruited at S’Anna Rehabilitation Center from January 2022 to June 2022. From an initial cohort of 96 TRK patients, we enrolled only those patients who met the following criteria: (1) were 55–80 years old; (2) had a unilateral TKR; (3) and their maximum time from surgery was 40 days. The exclusion criteria were (1) inflammatory disease after a TKR, and (2) any neurological, musculoskeletal, or other conditions affecting their movement. The criteria for enrollment have been adapted from previous studies [18,24]. In particular, considering only patients ranging from 55 to 80 years old is motivated by the high prevalence of OA in this age category [1,2]. Again, we included only patients with a unilateral TKR, to exclude bias due to a bilateral TKR. Finally, we considered only patients in the post-acute phase of the rehabilitation setting.

All the participants gave their written informed consent. The study was approved by the Ethical Committee of Regione Calabria (n° 169; 20 July 2019; Clinical Trial Number ISRCTN53840608 https://doi.org/10.1186/ISRCTN53840608 accessed on 23 September 2022), according to the Declaration of Helsinki.

### 2.2. Clinical and Kinematic Assessment

A clinical evaluation was made with the Numeric Rating Scale for Pain (NRS), which is a single 11-point numeric unidimensional measure of pain intensity in adults, [25] and the Barthel index (BI) to assess the patient’s activities of daily living following surgery [26]. The quantitative examination of the walking performance was carried out using the well-validated G-WALK wearable system [11] (BTS S.p.A.—Italy). An excellent intertrial reliability (ICC values between 0.84 and 0.99) and the concurrent validity for the speed, cadence, stride length, and stride duration are all evidenced in the gait characteristics recorded by the G-WALK sensor system [27]. Using ICC, which expressed a value between 0.799 and 0.977 between the consecutive measurements conducted in five days in total terms of the gait parameters, the G-Walk system exhibits a great reliability in the evaluation of the space–time of the gait parameters and angles of the pelvic girdle [28]. This system consists of a single 37 g miniaturized device that is attached to the subject’s L5 spinal segment using a semi-elastic belt. This device uses 4-sensor fusion technology and includes a triaxial accelerometer (16 bits/axis; 8 g), a triaxial magnetometer (13 bits; 1200 T), a triaxial gyroscope (16 bits/axes; 250°/s), and a GPS receiver. This enables the measurement of the gait, balance, symmetry, spatial orientation, and pelvic kinematics in space and time. The device is positioned on the patient’s waist utilizing an elastic belt on the 4th–5th lumbar vertebra, acquiring the acceleration values for the three anatomical axes [29,30]. Patients walk for 7 m, at a self-selected pace, as naturally as possible. The accelerometer’s weight was 62 g and contained a triaxial accelerometer (1000 Hz), triaxial gyroscope (8000 Hz), a triaxial magnetometer (100 Hz), and a GPS receiver (10 Hz). The G-WALK system also includes ProKin 252 and Walker View 3.0 SCX tools (Tecnobody, Dalmine (BG), Italy). Walker View 3.0 SCX is a treadmill featuring a 3D motion capture camera and an 8-load cell detecting surface. ProKin 252 is a portable platform controlled by an electropneumatic system for the proprioceptive-stabilometric assessment of both mono and bipodalic (Figure 1).

The registered data were transferred via Bluetooth to a laptop and analyzed utilizing a dedicated software (G-Studio, BTS Bioengineering S.p.A., Garbagnate Milanese (MI), 20024, Italy) to obtain the gait parameters, including walking for the kinematic parameters (stride length %, gait speed, cadence, stance and swing phase, single support, and double support phase), index of symmetry, propulsion index, and pelvis kinematics.

### 2.3. Study Design

A single-blind, randomized controlled study was conducted at the S’anna Institute, Crotone, Italy. The first stage was based on the recruitment of patients for the study. Next, eligible individuals underwent a clinical and kinematic examination at the baseline (T0). In the third stage, the participants were randomly assigned to two groups using a computer-generated randomization code. The following people were blinded to the patients’ group membership: the physicians (who carried out the clinical baseline assessment (T0) and post-treatment investigation (T1)) and the data entry assistants. In the fourth stage, the participants underwent experimental or traditional treatments. At the end of the treatment, the participants from both groups were given a final evaluation (T1), using the same protocol as the baseline.

### 2.4. Intervention

The Experimental group underwent a VFB group rehabilitation plan of 20 min sessions, 2 per day for 5 days a week, for six consecutive weeks. This treatment was combined with an additional 40 min of conventional therapy on the same days. The control group received only the conventional treatment (1 h session, 2/day for 5 days a week, for six consecutive weeks). The standard care rehabilitation for a TKR included exercises for: (a) ice/compression therapy; (b) the isometric contraction of operated leg muscles (especially in the first phase of the rehabilitation plan); (c) moderate muscular resistance training with a progressive load placed at the calf; and (d) active/passive knee joint mobilization with a physiotherapist’s assistance to enhance the knee’s range of motion. When the patients were able to bear their weight, they began gait and balance training in order to gradually become independent of the crutches (Figure 2).

The VBF plan was made using Walker View 3.0 SCX and ProKin 252 (Tecnobody, Bergamo, Italy), respectively, for gait training and balance training (Figure 3). Walker View 3.0 SCX is a sensorized treadmill with a 3-D camera that provides an immediate and objective dynamic image of the patient’s posture during training, the load of the lower limbs, the ROM of the requested joint, and the bending of the front, back, and side. ProKin 252 is an electro-pneumatic platform that, after an initial assessment phase, proposes a series of exercises that can be performed in the mono or bi-podalic mode with the biofeedback of the joint’s ROM and the center of the pressure. These tools include a series of rehabilitation games to maintain the patient’s concentration and improve their compliance to the therapy.

### 2.5. Statistical Analysis

A statistical analysis was performed using the Statistical Package for Social Science software (SPSS, v20.0, Chicago, IL, USA). Considering the small sample size, the non-parametric exact tests were used for the statistical analysis (Mann–Whitney exact test and the Wilcoxon exact test). The experimental and control groups were compared by the Mann–Whitney exact test. For both groups, the baseline [T0] was compared with the post-treatment [T1] by Wilcoxon exact test. All the statistical analyses were 2-tailed; the α levels were corrected for multiple comparisons using hierarchical Sidak correction. The level of significance was set at *p* ≤ 0.005.

A statistical power analysis was performed for the estimation of the sample size (G*Power, 3.1; https://www.psychologie.hhu.de (accessed on 1 January 2022). Considering our clinical experience together with the well-known reported effects of traditional interventions, we should expect an average increase of 45–50 points in the Barthel index clinical score after rehabilitation. From the standardization of the measures, and given the strong homogeneity of the sample, we expected a standard deviation of 40 for the total score. Therefore, with alpha = 0.005 and power = 0.9, the proposed sample size of 40 should be adequate for the main objective of this study. As a measure of the effect sizes, Cohen’s *d* was calculated, which indicates the magnitude of mean differences in the SD units.

## 3. Results

After the first screening, forty TKR patients fully met the admission criteria and were enrolled in the present study. The patients were randomly assigned to either the experimental group (VFB treatment) or the control group (Traditional treatment). Forty patients completed all phases of the rehabilitation protocol and were, finally, included in the statistical analysis (Figure 4).

At the time of inclusion, the two groups were perfectly matched for all the demographics (experimental group: 14 females and 6 males; control group: 13 females and 7 males; Chi2: 0.114, *p*-value: 0.74) and clinical variables (see Table 1). At the baseline, there were no significant differences in any demographic and clinical variables at the baseline between the two groups.

### Visual Feedback versus Traditional Treatments

Both groups showed a significant but similar improvement after rehabilitation, as measured by the clinical scales (Table 2) (NRS: the experimental group reduced their impairment from 63.5 ± 19.8 to 16.5 ± 10.4; the control group from 62.5 ± 12.1 to 16.5 ± 5.9; Barthel index: the experimental group improved their impairment from 46.75 ± 17.49 to 95.50 ± 6.67; the control group from 47.25 ± 6.78 to 95.35 ± 3.87, *p* < 0.001).

On the other hand, the gait analysis revealed a significant improvement in the experimental group with respect to the control group after the rehabilitation (Table 2). Indeed, after the VFB treatment, the TKR patients showed a significant improvement with respect to the control group in three gait indices: (a) the symmetry index (*p* < 0.001; *d* = 0.96); (b) single stance support (*p* < 0.0001; *d* = 2.59); and (c) obliquity (*p* < 0.0001; *d* = 1.45) (Figure 5).

## 4. Discussion

This randomized controlled study explored the efficacy of a new rehabilitation approach based on visual feedback for patients after a TKR compared with a standard rehabilitation. To the best of our knowledge, this is the first study applying this kind of rehabilitation approach (integrating VFB) on patients after a TKR, using gait analysis as an objective measurement of the clinical recovery. Overall, we found that the VFB intervention did not provide a significant gain in motor scores as measured by the standard clinical scales (Barthel index, NRS). However, the objective kinematic evaluation of the gait demonstrated that after the treatment, the patients were receiving a TKR gain of more than 10 points, as recorded by the parameters: (a) the symmetry index; (b) single stance Support; and (c) obliquity with respect to the control group.

The detected significant improvement in the symmetry index after the VFB treatment is in agreement with the findings of Zhang et al. [18] and Christensen et al. [31]. These authors found, respectively, an improvement in the Tinetti clinical score in postoperative patients with a knee fracture at 8 weeks after treatment, and an immediate improvement in the gait characteristics (interlimb symmetry) during a high-demand walking task in a patient after a TKR [18]. The symmetry index is one of the most important parameters to measure the gait and kinematic pattern alterations [32,33]. The symmetry index comprehensively estimates whether the operated limb and non-operated limb execute the corresponding gait cycle symmetrically, considering the stance and flight phase length [32]. The improved performance seen in the experimental group points to a more balanced distribution of the burden between the operated and unoperated limbs during walking, as well as a normalization of the gait cycle. Another well-known kinematic measure for assessing the locomotion is single stance support. The interval between the two footprints from the same foot, normalized for the pace time and measured as a percentage of the gait cycle [4] on the operated limb suggests that the soft tissue surrounding the knee has healed properly and that the patient is able to bear their weight properly on the operated limb [4]. Finally, our patients who received VFB showed better pelvic kinematics and more obliquity symmetry during walking. One of the basic systems controlling the gait mechanics is the pelvis [34]; the proper neuroactivation of the gluteal muscle and lower limb kinematics ensures a physiological movement (full knee extension) [35].

It is still debated that in the post-operative TKR, the damaging of the proprioceptors and the changes in the articular structure can further affect the lower limb stability and balance function, already altered by OA, resulting in a loss of the precise control of the effector because the bilateral proprioceptive receptor’s afferent information is asymmetrical [18]. Some authors point out that the model of prosthesis chosen has an influence on the proprioception, due to the alteration of the articular environment and structures as the cruciate ligament, meniscus, and intracapsular mechanoreceptors, that are essential in providing proprioceptive reflexes [36]. On the other hand, the importance of proprioceptive training in the early stage after a TKR is of growing interest for an advanced joint position sense and static and dynamic sense of movement. Proprioception has a conscious component (joint movement sense) and an unconscious component (postural and perturbation control) [37]; the noteworthy principle of the VFB is to link both components, visualizing the postural compensation, in order to achieve a patient’s self-correction of their own deficit. In this active learning, the patients integrated the multi-sensory information coordinating the peripheral signals, thanks to the help of vision [18]. Compared with the study conducted by Zhang et al. [18], our study gives a new insight, highlighting the efficacy of VFB even in patients with chronic OA, whit structures, and negative changes during the gait.

In the pre-intervention gait analysis, both patient groups displayed a substantial change in the gait metrics, including a decrease in cadence, velocity, cycle duration, cycle length, single stance support, propulsion, and symmetry index. In addition, we observed asymmetry in the tilt, obliquity, and rotation movements of the pelvis, indicating poor pelvis kinematics. These findings are consistent with those made by Bczkowicz et al. [4] and show a typical gait modification that reflects the patients’ post-TKR approach to reduce the knee moments and load on the painful articulation, which is quite prevalent in patients with high-grade OA. In particular, an unequal single support time and a reduced load on the operated limb, which are often linked to a worse knee function, are related to the reduced velocity after a TKR as compared to the normal range (decreased extension and muscle weakness) [4,38]. This could give new insight into the management of patients with TKR, confirming, in accordance with the finding of Levinger et al. [3], that the abnormal gait in the post-operative phase is a consequence of both the surgical intervention and persistence of pre-surgery walking patterns.

### Limitations

The main limitation of this study is the small sample size, which may have accounted for the lack of significant improvement shown by the experimental group in clinical standard evaluations (Barthel index/BRS). A further analysis with larger samples is needed to extend the application of the VFB approach to younger individuals.

## 5. Conclusions

Our study supports the need for a structured rehabilitation plan in TKR patients, that should be started in the early stage of the post-surgery period, to avoid the risk of not fully restoring gait biomechanics [3,4]. The rehabilitation protocol should not focus only on the knee function but should have a comprehensive insight about the patient’ recovery, integrating a gait retraining to optimize the recovery whilst avoiding compensation strategies [3]. The VFB treatment showed interesting results in improving the kinematic outcomes and should be considered in the rehabilitation of patients with TKR. Additionally, due to the high availability of this technology, in the future it could be incorporated into a telerehabilitation setting, allowing for the continuation of patient follow-up even after their discharge and ensuring that an exercise protocol is specifically designed for each patient in order to address their deficit and lower their risk of falling.

In conclusion, the use of gait analysis could support the physician to plan a rehabilitation protocol tailored to patient deficits.

## Figures and Tables

**Figure 1 jcm-11-07355-f001:**
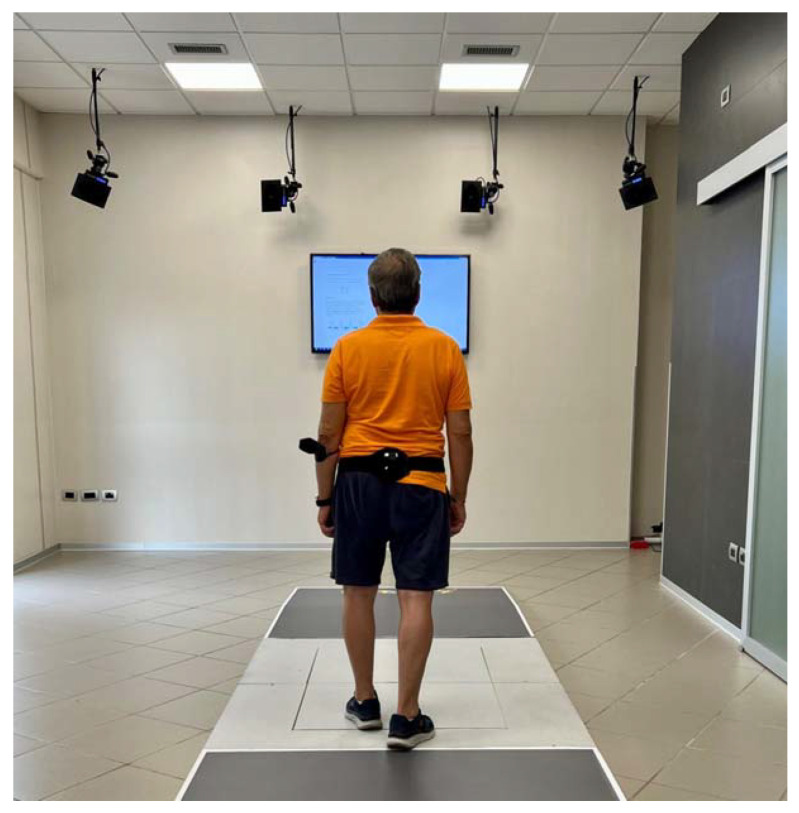
Gait analysis setting. The inertial measurement unit (G-Sensor, BTS bioengineering) is located on the 5th lumbar vertebra for a quantitative gait evaluation. Patients were asked to walk for 7 m. Data are elaborated with the manufacturer software (G-Studio, BTS bioengineering).

**Figure 2 jcm-11-07355-f002:**
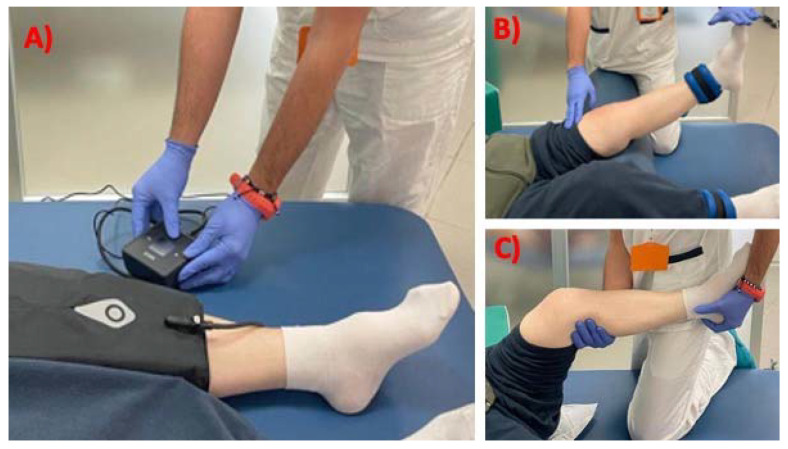
Standard care rehabilitation exercises for patients after total knee replacement, includes but not limited to: (**A**) Ice/compression therapy; (**B**) active resistance training with load at calf; (**C**) passive joint mobilization.

**Figure 3 jcm-11-07355-f003:**
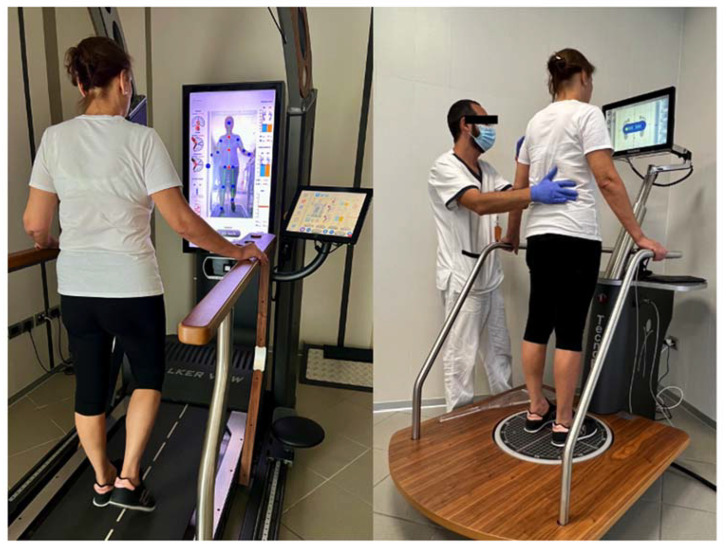
Tecnobody system for knee rehabilitation. Walker View 3.0 SCX tool (**left side**) and the ProKin 252 tool (**right side**).

**Figure 4 jcm-11-07355-f004:**
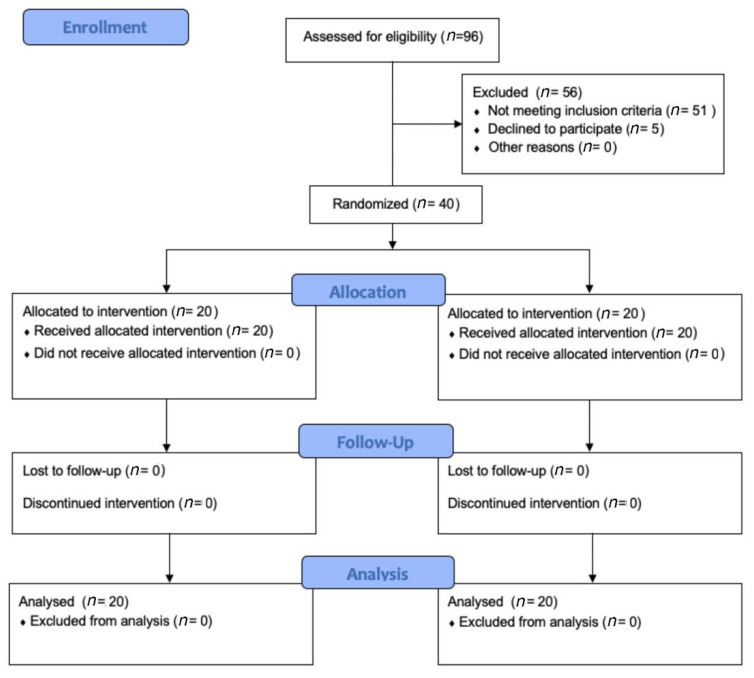
CONSORT Flow diagram showing the phases of a parallel randomized trial of two groups of TKR patients undergoing VFB (experimental group) or traditional (control) interventions.

**Figure 5 jcm-11-07355-f005:**
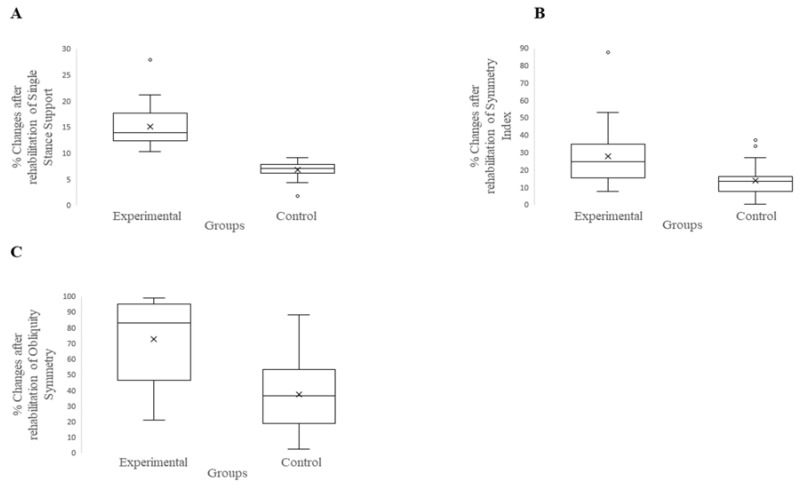
Boxplot of between groups post-intervention comparison on kinematic outcomes. (**A**): Single Stance Support; (**B**): Symmetry Index; (**C**) Obliquity Symmetry.

**Table 1 jcm-11-07355-t001:** Demographic and clinical characteristics at baseline.

	Experimental Group (n = 20)		Control Group (n = 20)		Between Groups Comparison			
	Mdn	Range	Mdn	Range	U	W	Z	*p*-Value
Age (yr)	70.5	66.3–76.5	71	69–73	196	406	−0.109	0.920
Height (cm)	161	160–164.5	160	159–161	153.5	363.5	−1.271	0.804
Body mass (kg)	67	62.5–69	66	64–67.75	172.5	382.5	−0.748	0.463
BMI	25.9	24.9–26.2	25.5	25–26.05	190.5	400.5	−0.257	0.804
Therapy duration (days)	37.5	33.3–39.8	36.5	31–40	191.5	401.5	−0.231	0.825

Mdn Median, Range Interquartile Range, U, Mann–Whitney U, W Wilcoxon W, BMI: body-mass index.

**Table 2 jcm-11-07355-t002:** Clinical and kinematic outcome measures in the experimental and control groups before and after treatments.

	Experimental Group					Control Group					Statistics between Groups at T1
	T0		T1			T0		T1			
	mdn	IQR	mdn	IQR	mdn Delta (%)	mdn	IQR	mdn	IQR	mdn Delta (%)	*p*-Value
Clinical Assessment											
NRS	60	50–77.5	10	10–20	−75%	60	52.5–70	20	10–20	−71.43%	0.278
Barthel index	40	35–63.75	100	95–100	118.75%	45	40–53.7	95	91–100	102.2%	0.381
Kinematic Gait Assessment											
Cadence	64.75	59.7–70	82.6	78.8–90.3	27.76%	61.1	48–80.4	82.35	76.4–89	30.37%	0.389
Velocity	0.56	0.5–0.7	0.8	0.7–0.8	39.59%	0.57	0.35–0.65	0.74	0.66–0.8	39.69%	0.465
Cycle duration	2.21	1.7–2.3	1.53	1.4–1.6	−28.51%	2.17	1.8–2.6	1.57	1.3–1.9	−27.37%	0.285
Cycle length	1.23	1.2–1.3	1.02	0.97–1.1	−12.57%	1.22	1.1–1.3	1.07	0.94–1.1	−12.79%	0.492
Symmetry index	72.7	63.5–77.9	87.15	84–92.8	24.64%	74.4	71.5–77.5	84	80.8–85.4	13.25%	0.001 *
Single stance support	32.88	31.9–34.1	37.8	36.6–38.9	13.89%	33.2	31.4–34.3	34.9	34.1–36.5	7.08%	<0.0001 *
Propulsion	3.3	2.73–3.8	4.8	4.25–6.1	55.59%	3.55	3.1–4.6	5.05	4.1–6.2	32.81%	0.007
Tilt symmetry	74.5	68.2–77.4	90.9	86.2–95	21.87%	77.3	64.3–83.2	93.25	77.4–95.6	18.33%	0.009
Obliquity Symmetry	42.65	36.5–50.1	73	62.3–82.1	83.16%	44.8	44.9–46.3	58.65	51.3–70.3	36.27%	<0.0001 *
Rotation symmetry	55.60	44.6–85.2	91.1	70.6–95	33.38%	63.5	61–66.5	76.55	73.9–78.7	20.01%	0.087

Data are shown as median and Interquartile range. statistical threshold * = *p* < 0.005. T0 = baseline; T1 = Follow-Up; NRS: Numeric Rating Scale for Pain; IQR = interquartile range.

## Data Availability

The dataset is available on request.

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
