# Peer review of "Efficacy of Visual Feedback Training for Motor Recovery in Post-Operative Subjects with Knee Replacement: A Randomized Controlled Trial"

_jcm, 2022, doi:10.3390/jcm11247355_

Round 1

Reviewer 1 Report

1.      The title should be changed to use the MDPI format, changing both the uppercase and lowercase characters.

2.      Please include all of the author’s emails after affiliation with name initials, except for the corresponding author based on MDPI format.

3.      The abstract requires the addition of quantitative results.

4.      Sort the keywords according to alphabetical order.

5.      Make the each of keywords with lowercase font following MDPI format, revise it.

6.      It is unclear whether the author's something new in this work. According to evaluation, several published studies by other researchers in the past adequately explain the issues you made in the present paper. Please be careful to highlight in the introduction section anything really innovative in this work.

7.      In order to demonstrate the research gaps that the current study aims to address, previous studies linked to it need to be explained in the introduction part, including their work, their novelty, and their limitations.

8.      Line 92, “…. In this study we…..:. It is not scientific form, make it into passive. Avoided the use of “we”.

9.      Gait data from patient is very essential that have been used to improving implant performance by study the materials and design via finite element model. The introduction and/or discussion part of an article should contain this crucial information. In addition, to support this explanation, the -suggested reference should be included as follows: Ammarullah, M. I.; Santoso, G.; Sugiharto, S.; Supriyono, T.; Kurdi, O.; Tauviqirrahman, M.; Winarni, T. I.; Jamari, J. Tresca Stress Study of CoCrMo-on-CoCrMo Bearings Based on Body Mass Index Using 2D Computational Model. Jurnal Tribologi 2022, 33, 31–8. https://jurnaltribologi.mytribos.org/v33/JT-33-31-38.pdf

10.   To help the reader grasp the study's workflow more easily, the authors could include more visuals to the materials and methods section in the form of figures rather than sticking with the text that now predominates.

11.   What is the basis for patient selection? Is there any protocol, standard, or basis that has been followed? It is unclear since the patient is very heterogeneous with a small number. The resonance involved impacts the present result makes this study flaws. One major reason for rejecting this paper.

12.   It is required to include additional information on tools, such as the manufacturer, the country, and the specification.

13.   Error and tolerance of experimental tools used in this work are important information that needs to be explained in the manuscript. It is would use as a valuable discussion due to different results in the further study by other researcher.

14.   Outcomes must be compared to similar past research.

15.   In the conclusion, please explain the further research.

16.   The reference should be enriched with literature from the last five years. Literature published by MDPI is strongly recommended.

17.   Due to grammatical problems and linguistic style, the authors should proofread the work. It would be used MDPI English editing service for this concern.

18.   Please be aware that the authors followed the MDPI format correctly; modify the current form and recheck, as well as any other problems that have been highlighted.

Author Response

  1. The title should be changed to use the MDPI format, changing both the uppercase and lowercase characters. 

Reply: Thank you for the comment. We changed the title following your indications.

  1. Please include all of the author’s emails after affiliation with name initials, except for the corresponding author based on MDPI format.

Reply: Thank you for the comment. We included the author’s name and initials

  1. The abstract requires the addition of quantitative results.

REPLY: Thank you for the comment. We improved the results section of the abstract as follows:

“Significant boosting of motor performance was detected in the experimental group with respect to the control group in the terms of symmetry index 84 (80.8-85.4) vs 87.15 (84-92.8) p = 0.001*; single stance support 34.9 (34.1-36.5) vs 37.8 (36.6-38.9); p<0.001; and obliquity parameters 58.65 (51.3-70.3) vs 73 (62.3-82.1); p<0.001.”

  1. Sort the keywords according to alphabetical order.

REPLY: We corrected the order of the keywords.

  1. Make the each of keywords with lowercase font following MDPI format, revise it.

REPLY: Done

  1. It is unclear whether the author's something new in this work. According to evaluation, several published studies by other researchers in the past adequately explain the issues you made in the present paper. Please be careful to highlight in the introduction section anything really innovative in this work.

  1. In order to demonstrate the research gaps that the current study aims to address, previous studies linked to it need to be explained in the introduction part, including their work, their novelty, and their limitations.

REPLY: We would like to thank this reviewer for these important suggestions. The introduction has been re-formulated to highlight the novelty of our work. 

In the last twenty years, much evidence has demonstrated the effectiveness of VFB for increasing the impact of gait recovery approaches[22], although this has been mostly applied to neurologic patients. Moreover, the VFB approach has been made through equipment not intended for rehabilitation (i.e. Wii) with limited clinical translation). Finally, another caveat is that the evaluation of motor gain in patients after TKR has largely been analyzed focusing on clinical (i.e WOMAC, VAS) or activity (TUG, FTSST, gait speed) scores [23].  For this reason, this study is aimed at evaluating, for the first time, the effectiveness of an advanced motor rehabilitation protocol integrating an innovative and user-friendly VFB system in patients after TKR through a detailed and objective kinematic gait analysis. For a deeper gait evaluation of patient outcomes (in terms of motor speed, balance, and control) an Inertial measurement unit (IMU) was also used.”

  1. Line 92, “…. In this study we…..:. It is not scientific form, make it into passive. Avoided the use of “we”.

REPLY: Done

  1. Gait data from patient is very essential that have been used to improving implant performance by study the materials and design via finite element model. The introduction and/or discussion part of an article should contain this crucial information. In addition, to support this explanation, the -suggested reference should be included as follows: Ammarullah, M. I.; Santoso, G.; Sugiharto, S.; Supriyono, T.; Kurdi, O.; Tauviqirrahman, M.; Winarni, T. I.; Jamari, J. Tresca Stress Study of CoCrMo-on-CoCrMo Bearings Based on Body Mass Index Using 2D Computational Model. Jurnal Tribologi 2022, 33, 31–8.  https://jurnaltribologi.mytribos.org/v33/JT-33-31-38.pdf

REPLY: Thank you for the suggestion. We improved the introduction section as follows, citing the articles you indicated.

“Gait analysis, instead, provides an objective assessment of motion pattern changes during walking in patients with TKR, as well as the improvement of implant function through the investigation of materials and design [4,14]”

  1. To help the reader grasp the study's workflow more easily, the authors could include more visuals to the materials and methods section in the form of figures rather than sticking with the text that now predominates.

REPLY: A new figure was added to the document to further illustrate our protocol. Please consider that this paper has 2 tables and 5 figures overall. These, in our opinion, adequately capture the key methodological aspects and clinical outcomes of our clinical study. 

  1. What is the basis for patient selection? Is there any protocol, standard, or basis that has been followed? It is unclear since the patient is very heterogeneous with a small number. The resonance involved impacts the present result makes this study flaws. One major reason for rejecting this paper.

REPLY: Thank you for your comment. We formulated this section: 

Criteria for enrollment have been adapted from previous studies [18,24,25]. In particular, considering only patients ranging from 55 to 80 years old is motivated by the high prevalence of OA in this age category [1,2]. Again, we included only patients with unilateral TKR, to exclude bias due to bilateral TKR. Finally, we considered only patients in the post-acute phase of the rehabilitation setting.”

  1. It is required to include additional information on tools, such as the manufacturer, the country, and the specification.

REPLY: Following the reviewer’s suggestion, we included this new statement: 

“The quantitative examination of walking performance was carried out using the well validated G-WALK wearable system [11](BTS S.p.A. - Italy). Excellent intertrial reliability (ICC values between .84 and .99), and concurrent validity for speed, cadence, stride length, and stride duration are all evidenced in the gait characteristics recorded by the G-WALK sensor system [...]. Using ICC, which expressed a value between 0.799 and 0.977 between consecutive measurements conducted in five days in total terms of gait parameters, the G-Walk system exhibits great reliability in the evaluation of space-time gait parameters and angles of the pelvic girdle [....]. This system consists of a single, 37-gram miniaturized device that is attached to the subject's L5 spinal segment using a semi-elastic belt. This device uses 4-sensor fusion technology and includes a triaxial accelerometer (16 bits/axis; 8g), a triaxial magnetometer (13 bits; 1200 T), a triaxial gyroscope (16 bits/axes; 250 °/s), and a GPS receiver. This enables the measurement of gait, balance, symmetry, spatial orientation, and pelvic kinematics in space and time. The device is positioned on the patient’s waist utilizing an elastic belt on the 4th-5th lumbar vertebra, acquiring the acceleration values for the three anatomical axes. [25,26]  Patients walk  for 7 meters, at a self-selected pace, more naturally possible. The accelerometer’s weight was 62 g and contained a triaxial accelerometer (1000 Hz), triaxial gyroscope (8000 Hz), a triaxial magnetometer (100 Hz) and a GPS receiver (10 Hz). The G-WALK system also includes ProKin 252 and Walker View 3.0 SCX tools (Tecnobody, Italy). Walker View 3.0 SCX is a treadmill featuring a 3D motion capture camera and an 8-load cell detecting surface. ProKin 252 is a portable platform controlled by an electropneumatic system for the proprioceptive-stabilometric assessment both mono and bipodalic (Figure 1).”

REPLY: 

  1. Error and tolerance of experimental tools used in this work are important information that needs to be explained in the manuscript. It is would use as a valuable discussion due to different results in the further study by other researcher.

REPLY: See reply above

  1. Outcomes must be compared to similar past research.

REPLY: Thank you for your comment. We improved the discussion as follows: 

Zhang et al. and Christensen et al. These authors found, respectively, an improvement in Tinetti clinical score in postoperative patients with knee fracture at 8 weeks after treatment, and an immediate improvement in gait characteristics (interlimb symmetry) during a high-demand walking task in patients after TKR. [18,32]

  1. In the conclusion, please explain the further research.

REPLY: Thank you for the comment. We explained as follows:

“Additionally, due to the high availability of this technology, in the future it could be incorporated into a telerehabilitation setting, allowing for the continuation of patient follow-up even after discharge and ensuring that an exercise protocol was specifically designed for each patient in order to address their deficit and lower their risk of falling.”

  1. The reference should be enriched with literature from the last five years. Literature published by MDPI is strongly recommended.

REPLY: 

Thank you for the comment, we enriched the references with recent articles.

  1. Pfeufer D, Gililland J, Böcker W, Kammerlander C, Anderson M, Krähenbühl N, Pelt C. Training with biofeedback devices improves clinical outcome compared to usual care in patients with unilateral TKA: a systematic review. Knee Surg Sports Traumatol Arthrosc. 2019 May;27(5):1611-1620. doi: 10.1007/s00167-018-5217-7. 
  2. Zhang T, Qui B, Liu HJ, Xu J, Xu DX, Wang ZY, Niu W. Effects of Visual Feedback During Balance Training on Knee Function and Balance Ability in Postoperative Patients After Knee Fracture: A Randomized Controlled Trial. J Rehabil Med. 2022 May 11;54:jrm00281. doi: 10.2340/jrm.v54.2209. PMID: 35322857; PMCID: PMC9131202.
  3. Marshall AN, Hertel J, Hart JM, Russell S, Saliba SA. Visual Biofeedback and Changes in Lower Extremity Kinematics in Individuals With Medial Knee Displacement. J Athl Train. 2020 Mar;55(3):255-264. doi: 10.4085/1062-6050-383-18.
  4. De Ridder R, Lebleu J, Willems T, De Blaiser C, Detrembleur C, Roosen P. Concurrent Validity of a Commercial Wireless Trunk Triaxial Accelerometer System for Gait Analysis. J Sport Rehabil. 2019 Aug 1;28(6):jsr.2018-0295. doi: 10.1123/jsr.2018-0295.
  5. Yazıcı MV, Çobanoğlu G, Yazıcı G. Test-retest reliability and minimal detectable change for wearable gait analysis system (G-Walk) measurements in children with cerebral palsy. Turk J Med Sci. 2022 Jun; 52 ( 3): 658-666. Doi: 10.55730 / 1300-0144.5358.
  6. Christensen JC, LaStayo PC, Marcus RL, Stoddard GJ, Bo Foreman K, Mizner RL, Peters CL, Pelt CE. Visual knee-kinetic biofeedback technique normalizes gait abnormalities during high-demand mobility after total knee arthroplasty. Knee. 2018 Jan;25(1):73-82. doi: 10.1016/j.knee.2017.11.010.

  1. Due to grammatical problems and linguistic style, the authors should proofread the work. It would be used MDPI English editing service for this concern.

REPLY: A complete English editing has been now performed

Reviewer 2 Report

The result of this study is interesting to researchers in this field. However, some information may improve the quality of the manuscript. My comments are as follows.

1.         The number of patients mentioned in abstract and results is different. Please check it again.

2.         It will be better if there is a more detailed explanation including figures regarding the exercises (Line 143-146) conducted in this study.

3.         It will be better if there is more detailed definition of the kinematics parameters (Line 119-121). Figures may help authors to easily understand the parameters.

4.         Figure 4 is too small and is not clear.

Author Response

The result of this study is interesting to researchers in this field. However, some information may improve the quality of the manuscript. My comments are as follows.

Thank you for the positive comment. 

  1. The number of patients mentioned in abstract and results is different. Please check it again.

REPLY: Thank you for the comment. The number of patients was 40, of which 27 females and 13 males were divided Experimental Group: 14 females and 6 males, and the Control Group: 13 females and 7 male

  1. It will be better if there is a more detailed explanation including figures regarding the exercises (Line 143-146) conducted in this study.

REPLY: Thank you for the comment. We improved the section as suggested and a new figure has been included.

“The standard care rehabilitation for TKR included exercises for: a) isometric contraction of operated leg mus­cles (especially in the first phase of the rehabilitation plan); b) moderate muscular resistance training with a progressive load placed at the calf; c) active/passive knee joint mobilization with a physiotherapist's assistance to enhance knee range of motion. When the patients were able to bear weight, they began gait and balance training in order to gradually become independent of the crutches” 

  1. It will be better if there is more detailed definition of the kinematics parameters (Line 119-121). Figures may help authors to easily understand the parameters.

REPLY: Figure 1 represents the correct setup of the IMU fixed through the elastic belt on L5-S1. However, following the reviewer’s suggestion, a new section has been added to better explain our kinematics evaluation. 

“The quantitative examination of walking performance was carried out using the well-validated G-WALK wearable system [11](BTS S.p.A. - Italy). Excellent intertrial reliability (ICC values between .84 and .99), and concurrent validity for speed, cadence, stride length, and stride duration are all evidenced in the gait characteristics recorded by the G-WALK sensor system [...]. Using ICC, which expressed a value between 0.799 and 0.977 between consecutive measurements conducted in five days in total terms of gait parameters, the G-Walk system exhibits great reliability in the evaluation of space-time gait parameters and angles of the pelvic girdle [....]. This system consists of a single, 37-gram miniaturized device that is attached to the subject's L5 spinal segment using a semi-elastic belt. This device uses 4-sensor fusion technology and includes a triaxial accelerometer (16 bits/axis; 8g), a triaxial magnetometer (13 bits; 1200 T), a triaxial gyroscope (16 bits/axes; 250 °/s), and a GPS receiver. This enables the measurement of gait, balance, symmetry, spatial orientation, and pelvic kinematics in space and time. The device is positioned on the patient’s waist utilizing an elastic belt on the 4th-5th lumbar vertebra, acquiring the acceleration values for the three anatomical axes. [25,26]  Patients walk for 7 meters, at a self-selected pace, more naturally possible. The accelerometer’s weight was 62 g and contained a triaxial accelerometer (1000 Hz), triaxial gyroscope (8000 Hz), a triaxial magnetometer (100 Hz) and a GPS receiver (10 Hz). The G-WALK system also includes ProKin 252 and Walker View 3.0 SCX tools (Tecnobody, Italy). Walker View 3.0 SCX is a treadmill featuring a 3D motion capture camera and an 8-load cell detecting surface. ProKin 252 is a portable platform controlled by an electropneumatic system for the proprioceptive-stabilometric assessment of both mono and bipodalic (Figure 1).”

  1. Figure 4 is too small and is not clear.

REPLY: Figure 4 has been modified

Reviewer 3 Report

A randomised control with a definite difference between the controls and treatment group is always desirable. However a few reservations. The  numbers in each limb are small. 

The rehab results are not so overwhelmingly better to mandate that this is a must use therapy in post op TKR patients. I cannot see my colleagues eager to prioritise this as part of our rehabilitation programme.

Overall of moderate interest but probably worthy of publication

Author Response

A randomised control with a definite difference between the controls and treatment group is always desirable. However a few reservations. The  numbers in each limb are small. 

The rehab results are not so overwhelmingly better to mandate that this is a must use therapy in post op TKR patients. I cannot see my colleagues eager to prioritise this as part of our rehabilitation programme. Overall of moderate interest but probably worthy of publication

REPLY: We would like to thank this reviewer for his valuable comments. Generally speaking, the innovation introduced by our paper lies in:

  1. a) the translation of the visual feedback approach for recovering patients after TKR. Indeed, the idea of visual feedback has widely been discussed in the literature, mostly for neurologic patients.
  2. b) the employment of a well-validated rehabilitation device (TecnoBody®). Generally, the VFB approach has been made through equipment not intended for rehabilitation (i.e. Wii) with limited technological translation into clinical practice.
  3. c) the employment of a well-validated objective evaluation of gait performance. In this field of study, another caveat is that the evaluation of motor gain in patients after TKR has largely been measured using clinical scales. For this reason, this study is aimed at evaluating, for the first time, the effectiveness of an advanced motor rehabilitation protocol integrating an innovative and user-friendly VFB system in patients after TKR through a detailed and objective kinematic gait analysis. For a deeper gait evaluation of patient outcomes (in terms of motor speed, balance, and control) an Inertial measurement unit (IMU) was also used.”

Round 2

Reviewer 1 Report

It is great work, well done.